# Application of Hyperspectral Imaging to Underwater Habitat Mapping, Southern Adriatic Sea

**DOI:** 10.3390/s19102261

**Published:** 2019-05-16

**Authors:** Federica Foglini, Valentina Grande, Fabio Marchese, Valentina A. Bracchi, Mariacristina Prampolini, Lorenzo Angeletti, Giorgio Castellan, Giovanni Chimienti, Ingrid M. Hansen, Magne Gudmundsen, Agostino N. Meroni, Alessandra Mercorella, Agostina Vertino, Fabio Badalamenti, Cesare Corselli, Ivar Erdal, Eleonora Martorelli, Alessandra Savini, Marco Taviani

**Affiliations:** 1National Research Council, Institute of Marine Sciences, Via Gobetti 101, 40129 Bologna, Italy; federica.foglini@bo.ismar.cnr.it (F.F.); mariacristina.prampolini@bo.ismar.cnr.it (M.P.); lorenzo.angeletti@bo.ismar.cnr.it (L.A.); giorgio.castellan@bo.ismar.cnr.it (G.C.); alessandra.mercorella@bo.ismar.cnr.it (A.M.); marco.taviani@bo.ismar.cnr.it (M.T.); 2Department of Earth and Environmental Sciences and CoNISMa LRU, University of Milano - Bicocca, Piazza della Scienza 1, 20126 Milano, Italy; fabio.marchese1@unimib.it (F.M.); valentina.bracchi@unimib.it (V.A.B.); a.meroni9@campus.unimib.it (A.N.M.); agostina.vertino@unimib.it (A.V.); cesare.corselli@unimib.it (C.C.); alessandra.savini@unimib.it (A.S.); 3Department for the Cultural Heritage, University of Bologna, Via degli Ariani 1, 48121 Ravenna, Italy; 4Department of Biology and CoNISMa LRU, University of Bari, Via Orabona 4, 70124 Bari, Italy; giovanni.chimienti@uniba.it; 5Ecotone AS, Pirsenteret Havnegata 9, inngang 1 – 3 etg, 7010 Trondheim, Norway; ingrid@ecotone.com (I.M.H); magne@ecotone.com (M.G.); ivar@ecotone.com (I.E.); 6Ghent University, Campus Ufo, Rectorate, Sint-Pietersnieuwstraat 25, B-9000 Ghent, Belgium; 7National Research Council, Institute for the Study of Anthropic Impacts and Sustainability in Marine EnvironmentsVia Giovanni da Verrazzano 17, 91014 Castellammare del Golfo, Trapani, Italy; fabio.badalamenti@cnr.it; 8National Research Council, Institute of Environmental Geology and Geoengineering, Via Salaria km 29,300, 00015 Monterotondo, Rome, Italy; eleonora.martorelli@uniroma1.it; 9Biology Department, Woods Hole Oceanographic Institution, 266 Woods Hole Road, Woods Hole, MA 02543, USA; 10Stazione Zoologica Anton Dohrn, Villa Comunale, 80121 Naples, Italy

**Keywords:** hyperspectral camera, spectral library, habitat mapping, coralligenous, cold-water coral, Adriatic Sea

## Abstract

Hyperspectral imagers enable the collection of high-resolution spectral images exploitable for the supervised classification of habitats and objects of interest (OOI). Although this is a well-established technology for the study of subaerial environments, Ecotone AS has developed an underwater hyperspectral imager (UHI) system to explore the properties of the seafloor. The aim of the project is to evaluate the potential of this instrument for mapping and monitoring benthic habitats in shallow and deep-water environments. For the first time, we tested this system at two sites in the Southern Adriatic Sea (Mediterranean Sea): the cold-water coral (CWC) habitat in the Bari Canyon and the Coralligenous habitat off Brindisi. We created a spectral library for each site, considering the different substrates and the main OOI reaching, where possible, the lower taxonomic rank. We applied the spectral angle mapper (SAM) supervised classification to map the areal extent of the Coralligenous and to recognize the major CWC habitat-formers. Despite some technical problems, the first results demonstrate the suitability of the UHI camera for habitat mapping and seabed monitoring, through the achievement of quantifiable and repeatable classifications.

## 1. Introduction

Traditionally, underwater habitat mapping has been carried out coupling acoustic remote sensing techniques with red/green/blue (RGB) images, videos and bottom sampling [1,2]. The analysis of video and images is performed manually by expert interpretation, or automatically when a photomosaic is available [3]. Recently, the European programs such as the EU Marine Strategy Framework Directive (MSFD: 2008/56/EC), require the monitoring of benthic habitat extent and distribution (Criteria 1.4 and 1.5 of the Descriptor 1 “Biological Diversity”). The MSFD scope is to assess the good environmental status (GES) of European water [4] with the lowest possible impact on the seafloor. This effort is translated into quantifiable operational indicators that should be measurable at different scale and repeatable in time [4]. To fulfill these requirements, there is a need for innovative approaches and tools to obtain detailed, reliable, quantifiable and repeatable maps of relevant habitats in different underwater environments [2]. 

During the last decade, the implementation of hyperspectral devices has become a viable alternative to regular photography. In contrast to ordinary cameras that acquire three colour bands (RGB), hyperspectral cameras record the full spectrum of reflected light, in each pixel of the acquired image. Therefore, the spectral resolution and amount of information obtainable from an image transect is vastly increased compared to traditional photography [5]. As a result, UHI can detect the subtle and otherwise unnoticeable spectral properties of a given OOI and record object-specific optical fingerprints. Optical fingerprinting increases classification accuracy for both qualitative and quantitative mapping [5].

This technology has previously been applied to airborne remote sensing, both in terrestrial and marine environments, through passive sensors requiring sunlight. However, sunlight is highly attenuated in marine waters [6,7]. As a consequence, this technique is suitable only in coastal areas and relatively shallow water (up to 50 m depth: [8,9,10]). Hence, works on hyperspectral imaging from satellite or airplanes are focussed on oceanographic and biological studies [11,12], mapping of ocean colour [12,13] and shallow benthic habitats [8,10] such as coral reefs [14,15,16,17], seagrasses [18,19,20] and kelp forests [9].

Recently, different instrument carriers for the underwater hyperspectral imager (UHI) have been used in underwater field applications, such as the customized scanning rig [21,22], remotely operated vehicle (ROV) [6,7,23,24,25,26,27] or autonomous underwater vehicle (AUV) [28]. The UHI has been tested and utilized for different purposes from shallow (< 6 m) [22] to abyssal depths (ca. 4200 m depth) [6,7]. Among the many applications, UHI was related to the identification of manganese nodules [6], infrastructure inspection, seafloor impact of offshore drilling [29] and marine archaeology [25,27,30]. However, the most reported UHI application is within the field of benthic habitat mapping, modelling and monitoring. Underwater hyperspectral imaging with ROV has been used to study coastal kelp forests [25], vertical rock wall habitat and soft sediments [24], red calcareous algae and associated fauna [25,26], deep-sea megafauna [7] and CWC communities [23,25]. Laboratory experiments to measure changes in the health status of CWCs exposed to hydrocarbons emissions is another application of UHI [31].

As a first application to the Mediterranean basin, we tested the UHI in the Adriatic Sea [32]. The semi-enclosed Adriatic Sea hosts a variety of benthic habitats, including the shallow oyster reefs and sponge communities in the Venice Lagoon [33], coralligenous formations on the shelf (e.g., [34] with references therein), down to the CWC habitat in deep water (> 200 m) in the south (e.g., [35,36,37,38,39,40]). At present, the Adriatic Sea is under siege by a number of stressors, such as high demographic pressure on its coastal areas, pollution, marine littering and dumping, fishing practices, ship traffic, harbour activities and industrial operations [41]. Our study targets the distribution and extent of two biogenic habitats (CWC and Coralligenous), in different geomorphological and depth contexts, considered to be of key importance in monitoring plans. In this perspective, the UHI may prove useful in habitat mapping to meet the requirements of European programs (e.g., EU MSFD). 

## 2. Materials and Methods

### 2.1. Study Area

The two selected sites are located in the Southern Adriatic Sea (Figure 1): the Bari Canyon (CWC habitat) and the continental shelf off-shore Brindisi (Coralligenous habitat). 

The Bari Canyon site is located ca. 40 km away from the city of Bari, on the continental margin, within a well-known Cold-Water Coral ecosystem, extending between −200 and −700 m on the southern flank of the canyon [35,36,38,42,43]. The CWC habitat is here characterised by complex megabenthic communities, mainly represented by the colonial scleractinian *Madrepora oculata,* subordinately *Desmophyllum pertusum* (*Lophelia pertusa* [44]) and the solitary *Desmophyllum dianthus*, and by large fan-shaped sponges (i.e. *Pachastrella monilifera* and *Poecillastra compressa*) [35,36,37,40]. 

The Brindisi site is placed on a flat continental shelf, about 10 km far from the coast at an average depth of 30 m. Coralligenous outcrops, mosaicking coarse biogenic sediments [34,35,36,37,38,39,40,41,42,43,44,45,46], dominate the seafloor. The coralligenous is a very complex habitat where crustose coralline algae (CCA) and red algae belonging to the order of Peyssonelliales are often the main bioconstructors in shallower waters, generating a new solid substrate and constituting a three-dimensional biogenic build-up [47,48,49,50,51,52]. It represents a key habitat of the Mediterranean continental shelf because of its structural and functional importance, as well as for its considerable aesthetic value [53]. In the study area, discrete coralligenous build-ups [46] characterize the seafloor, with a thickness up to 70 cm. CCA, usually growing in dim light conditions, and other algae such as Peyssonelliales primarily form these solid substrates; bryozoans and serpulids contribute to the bioconstruction [54,55]. Moreover, these hard substrates host different fauna and flora, often overgrowing the calcified red algae [51]. 

### 2.2. Underwater Hyperspectral Imager (UHI)

The underwater hyperspectral imager (UHI), developed and patented by Ecotone AS, consists of a waterproof housing containing camera system, computer and data storage. It is operated with a light source for proper illumination, and represents a new system for the identification, mapping and monitoring of OOI at the seabed [5,25].

The UHI is a push broom camera, which records one line at time. It is equipped with a spectrograph that receives light through a thin entrance slit [57]. Light entering the instrument is diffracted into separate wavelengths and projected onto the camera sensor as a contiguous spectrum. A continuous image is built up line by line and can be presented on a monitor for visualisation [26]. 

The UHI needs to be mounted on a moving platform (e.g., ROV) that operates at a fairly constant speed and altitude from the seafloor. Mobile platforms equipped with dynamic positioning systems permit larger areal coverage, and the possibility of re-visiting the surveyed sites based on their geolocation data [58]. External lamps provide seafloor illumination. Due to the rapid attenuation of light in the marine environment, UHI is normally confined to scanning altitudes <5 m above the OOI, depending on water quality and turbidity. 

Image lines are captured perpendicular to the direction of movement. The result is a hyperspectral image, featuring detected intensities for all the wavelengths used [57]. The spatial resolution provided by hyperspectral cameras varies with altitude, exposure time of the camera and speed vessel.

### 2.3. Data Acquisition

In February 2017, we carried out the SPECTRA17 cruise on board of the R/V Minerva Uno, aimed at testing the ability of the UHI to acquire seafloor hyperspectral images in the Southern Adriatic Sea. 

An Ecotone UHI (Model 4) was mounted vertically beneath the ROV Super Mohawk II 34 Observation Class, facing directly towards the seafloor, together with two LED lamps (3200 lumens per lamp) oriented at 90°, a 2D high resolution RGB camera with two lamps oriented at 60° and a system to correct UHI camera motion for pitch, roll and yaw. Acquisition and pre-processing of data were managed through the C++ based UHI customized software Immersion installed on the topside control unit.

The ROV maintained constant speed (about 1 knot) and heading in each dive (45° for CWC and 25° for coralligenous, respectively), at a constant altitude of 1–1.5 m. The track length was about 2.5 km for the CWC site and 785 m for the coralligenous site. 

ROV navigation and position was provided by Low-Accuracy TrackLink USBL Positioning System (accuracy of <2% of the water depth), positioning data were recorded at 1 sec. The positioning system was operated using the software PDS2000. In addition to UHI, a black and white camera for ROV navigation and a high-resolution camera provided footage for manual identification of OOI. 

We reoccupied previous ROV tracks, making use of high-resolution videos and photos already available for comparing the UHI results and classification.

### 2.4. High Resolution Camera Image Data Processing

The RGB images for coralligenous and CWC sites, collected by the high-resolution camera, were processed using ADELIE Software by IFREMER and used to identify and classify the OOI. ADELIE is a software able to synchronise video and navigation and then automatically capture georeferenced still images (or image sequences) for a chosen time interval (e.g., every 10 s). Through a specific module of ADELIE based on ArcGIS Desktop, it is possible to filter and smooth vehicle navigation, to have direct access to pictures and to localize video in real time. Following this procedure, we mapped all track lines producing benthic habitat maps and estimated the extent of habitat at ROV scale.

### 2.5. UHI Data Processing

Processing of hyperspectral images consisted of three main steps: (1) Radiometric processing correcting for sensor influence; (2) Georeferencing to assign geospatial information and perform image geocorrection; (3) Conversion of radiance to reflectance by correction of external influences from illumination source. Using Immersion, we performed the georeferencing and radiometric correction of the acquired images to produce non-distorted and georeferenced hyperspectral images. The spectral resolution is up to 2.2 nm, while the spatial resolution is 1 cm for CWCs and 0.5 cm for coralligenous. The resulting UHI coverage is about 1–1.2 m width, depending on height above seafloor. The pre-processed UHI images used in this work are available as Research Object (for further details see Appendix A).

#### 2.5.1. Radiometric Processing

The UHI was calibrated in the laboratory before the oceanographic mission. A standard lamp with known spectral properties was used to find the ratio between observed the digital counts of intensity for each wave band on the sensor, to spectral radiance (W m^−2^ sr^−1^ nm^−1^). These measurements can be applied on raw recordings from the field to correct for sensor-specific noise and dark current, as well as data acquisition parameters such as exposure time and binning. Radiance conversion was automatically performed as part of the georeferencing algorithm in Immersion.

#### 2.5.2. Georeferencing

Geographic position and spatial correction of the hyperspectral images were provided by the georeferencing procedure through Immersion software by using: (1) USBL data for the ROV position and (2) altitude data from the Ecotone IMU (Inertial Measurement Unit). As the navigation produced by USBL track link contains frequent spikes and some metrical gaps, we statistically filtered navigation and altitude data through 20- and 5-point-wide windows, respectively, using an adjacent averaging smoothing algorithm to improve resolution [59]. In addition to navigation, motion and altitude data the following parameters for Immersion were set: (1) course-made-good option for ROV heading calculation; (2) a spectral binning of 8 resulting in 28 bands with 15 nm resolution; (3) a spatial binning of 1 and (4) a cell resolution of 1 cm for CWCs and 0.5 cm for coralligenous.

#### 2.5.3. Reflectance Processing

Following the procedure in [1], radiance data was converted into reflectance correcting for the external influence on the spectral characteristics from LED lamps or their combination with sunlight, at the deep and shallow site, respectively. These spectral characteristics are not definable, so we approximated the illumination influence by using a reference spectrum calculated for the entire analysed segment of both sites in R software [60]. Then, we divided each image pixel by its respective reference spectrum. 

### 2.6. UHI Spectral Supervised Classification

We selected a segment from the CWC track line of about 10 m and a segment from the coralligenous track line of about 7.5 m for the spectral supervised classification and further analysis. Classification was performed through the software ENVI 5.5, using the spectral angle mapper (SAM) method. The accuracy was determined generating a confusion matrix for each site. 

#### 2.6.1. Spectral Angle Mapper (SAM)

The spectral angle mapper (SAM) is a supervised classification technique that measures the similarity of image spectra to reference spectra. The reference spectra can be measured in field or laboratory, they can be taken directly from the image as region of interest (ROI) or imported from already known spectral libraries. SAM measures similarity by calculating the angle between the two spectra, treating them as vectors in n-dimensional space, with n being the number of bands [61]. The angle is the arccosine of the dot product of the two spectra. Smaller values for the angle indicate higher similarity between pixel and reference spectra. As the angle between two vectors is independent of the vector length, this method is unaffected by gain factors, such as solar illumination [62]. The SAM only compares the angle between the spectral directions of the reference and test pixels considered, there is no specific requirement for a large amount of training samples [63].

We trained the model selecting ROIs that include spectral signatures representing substrates, megaflora and megafauna (> 2 cm) present in the surveyed areas. The selected ROIs reflect the highest spectral diversity due to different pigmentation of the OOI and make up the benthic classes that will constitute the final classification. 

In particular for the CWC site, we used 10 ROIs to train the SAM classification representing 4 benthic classes: (1) colonial cnidarian, (2) sponge, (3) mud, (4) bedrock (Table 1). For colonial cnidarian, bedrock and sponge we selected multiple ROIs due to the differences of the UHI RGB colours along the segment. For the coralligenous site, we created 13 ROIs identifying 5 benthic classes: (1) CCA and Peyssonelliales (P) forming the build-ups (CCA+P); (2) green algae on build-ups (in particular *Codium bursa* and *Flabellia petiolata*) and on sediment (*Caulerpa prolifera*); (3) Seagrass (*Posidonia oceanica*); (4) organisms associated with the presence of build-ups; and (5) sand (Table 2). For the CCA+P and the green algae (Green algae 1 and 2) classes, we selected more than one ROI due to the illumination unevenness caused by the slight altitude variation along the track.

After running the SAM with a maximum angle of 0.1 radiant, we applied the ENVI ‘Rule classifier’ post classification tool to adjust the threshold angles for each class and improve the classification results. We choose the appropriate thresholds classifying by minimum values and visualizing the histogram that shows the frequency of pixel with different angles. Finally, we used the ENVI ‘Majority/Minority analysis’ tool with a kernel size of 3 × 3 to clean and smooth the SAM classification. 

#### 2.6.2. SAM Classification Accuracy

We generated a confusion matrix for both test sites using ENVI and ArcGIS desktop to determine the accuracy of the classification results. Firstly, we produced eight random sampling points for each benthic class using the ‘Generate random sample’ ENVI tool with the equalized random technique to divide the population into homogeneous subgroups, while ensuring that each class sample size was the same (1 pixel). Then, we compared within ArcGIS Desktop the random sampling points predicted classes with the UHI image (ground truthing), to assign the real class for each point as defined by the expert interpretation. We generated the confusion matrix for each classification using the predicted classes and the real class values specifying the overall, producer’s and user’s accuracies. The overall accuracy is calculated by summing the number of correctly classified values and dividing by the total number of values. The user’s accuracy (UA) is the number of correctly identified pixels in a class, divided by the total number of pixels of the class in the classified image; it shows false positives, where pixels are incorrectly classified as a known class when they should have been classified as something else. Producer’s accuracy (PA) is the number of correctly identified pixels divided by the total number of pixels in the reference image; it gives a false negative, where pixels of a known class are classified as something other than that particular class.

## 3. Results

### 3.1. Spectral Library for CWC Site

The mean spectra of the 10 ROIs selected for the CWC site are shown in Figure 2. Mud and bedrock have an almost constant level of reflectance along all wavelengths. The bedrock shows a deflection point at 473 nm, with a minimum at 500 nm (Figure 2A,B). The sponge 1 has an inflection point at 500 nm and an increasing trend between 555 and 680 nm (from green to red) (Figure 2C). Sponge 2 and 3 display similar patterns with a dissimilar level of reflectance due to different illumination, an inflection point at 530 nm and highest values in the orange/red part of the spectrum between 630 nm and 670 nm. Sponge 4 has a slightly different shape and a smoothed slope, possibly caused by a minor colour difference. Despite all sponges seem to pertain to the same morphotype (belonging to the white/orange large fan-shaped *P. compressa* and/or *P. monilifera*), it is not possible to attribute the accurate species to each ROI and to each spectrum (Figure 2D). The colonial cnidarian 2 and 3 have a spectral signature with a similar pattern, constant along the entire wavelength range, with a difference in the level of reflectance, while colonial cnidarian 1 shows a slight peak in the blue part of the spectrum (about 470 nm) (Figure 2E). According to previous studies (e.g., [35,36,38,43,56]), we can assume that the most probable species belong to *M. oculata*, while the *D. pertusum* appears to be rarer in this site.

For benthic classes with multiple ROIs (colonial cnidarian, sponge and bedrock), we analyzed the spectral differences and considered the mean for spectra with the same pattern but a different reflectance intensity (Figure 2F). The three ROIs of the colonial cnidarians are recognized as a unique class, because the peak in the blue part of the spectrum for colonial cnidarian 1 is considered an artefact. 

### 3.2. Spectral Library for the Coralligenous Site

The mean spectra of the 13 ROIs selected for the coralligenous site are shown in Figure 3. The CCA+P1, CCA+P2, CCA+P3 have the inflection point at 600 nm with a maximum between 630 and 670 nm in the red part of the spectrum. The wide range of standard deviation probably reflects the high biodiversity of this category. We can assume that the three spectra, with a similar pattern and a different level of reflectance intensity, belong to the same benthic class CCA+P.

The green algae on build-ups (green algae 1+2), mainly represented by *C. bursa* and *F. petiolata*, have a maximum at about 515 nm (green part of the spectrum), while the green algae *C. prolifera* (Green algae 3) and the seagrass *P. oceanica* show a slight difference with a maximum of reflectance at 595 and 528 nm, respectively. The ascidian *Halocynthia papillosa* and the starfish *Echinaster sepositus* show a peak at 650 and 680 nm in the red part of the spectrum. The serpulids show an almost constant spectrum with a small inflection at 610 nm. The two species belonging to the genus *Axinella* have similar spectra with a maximum between 630 and 670 nm (red part of the spectrum). The sand has a constant value of reflectance along all wavelengths.

### 3.3. Supervised Classification Results for CWC Site

Based upon the previous considerations about the spectra, we can define six benthic classes for the final classification: (1) colonial cnidarian, (2) sponge 1, (3) sponge 2+3, (4) sponge 4, (5) mud and (6) bedrock (Figure 4). The mud class results were the most dominant (40%) followed by the bedrock (23.6%). The colonial cnidarian class is scattered along the track with a total coverage of 0.4%, the sponge classes (sponge 1, sponge 2+3, sponge 4) are patchy with a total coverage of 0.3%. The illumination problems, due to low lamp potential, ROV orientation and dipping of the substrate (the canyon flank in this site is up to 25°), cause a homogenous dark shading on the right (deeper) section of the image. This results in a high percentage of unclassified pixels (26.5%), also including colonial cnidarians and sponges still visible on the RGB image. However, these technical issues do not prevent the clear discrimination of the benthic classes and the estimation of percentage cover.

### 3.4. Supervised Classification Results for Coralligenous Site

For the coralligenous we defined 10 benthic classes for the final classification (Figure 5), according to the analysis of spectral signatures: (1) CCA+P, (2) green algae 1+2, (3) *C. prolifera*, (4) *P. oceanica*, (5) *Axinella* sp. 1, (6) *Axinella* sp. 2, (7) serpulids, (8) *E. sepositus*, (9) *H. papillosa*, (10) Sand. The sand class is the most dominant in the area with a coverage of 60%. The build-ups are generally well discriminated along the entire track with a coverage of 29% for CCA+P and 8% for the green algae on build-ups (Green algae 1+2). The SAM identifies well green algae *C. prolifera* (Green algae 3) and the seagrass *P. oceanica* with a coverage of 0.7% and 0.9% of the total classified area, respectively. The taxa associated with coralligenous are easily distinguishable with a total coverage of 0.13%. We highlight the low percentage of unclassified pixel (1.5% in total). 

### 3.5. Classification accuracy for CWC and Coralligenous Sites

The CWC SAM has a high overall accuracy (84.38%) as determined from the confusion matrix comparing the predicted classes with the real classes (see Table 3). The colonial cnidarian class gives 100% UA and 87.5% PU showing the presence of a minimum number of false negative along the deepest section of the analysed segment. The sponges (sponge 1, sponge 2+3, sponge 4) show a lower value in terms of PA (87.50%) than UA (100%) due to the presence of false negative. The bedrock and mud classes show almost the same accuracy (>60% both for PA and UA) both for the presence of false negative and positive. 

The overall accuracy of the SAM for the coralligenous is 72% (see Table 4). There is a high discrepancy between PA and UA for the *P. oceanica* and *C. prolifera* (Green algae 3) classes with no commission errors (100% UA), but with a level of reliability of 37.5% and 50% due to a higher omission errors. For the sand class, the UA is extremely low (25%) because of the high number of false positive. For the CCA+P both PA and UA are 100%. The green algae on build-ups (green algae 1+2) shows no false positives (100% UA) and a PA higher than the overall value (87.5%). In general, all organisms associated with the coralligenous build-ups have a high level of accuracy (> 88% both for PA and UA), with the exception of *Axinella* sp. 1 (75% PA) showing the presence of false negatives. 

## 4. Discussion

### 4.1. Evaluation of the Acquisition Set-up and Suggestion of Best Practice for Data Collection

The acquisition of high-quality UHI is challenging because of the requirements to perform a satisfactory survey, such as maintaining a constant speed, heading and altitude, as well as high-resolution navigation data [5,6,7]. Proper illumination is mandatory to avoid the acquisition of “striped” images, characterised by shadowed and hyper-illuminated areas, leading to misclassification of the acquired images.

In our survey, the ROV met most requirements, yet seafloor illumination and navigation data were not optimal. At both sites, the illumination was uneven due to a non-optimal configuration of the lamps (two lamps for the ROV RGB camera running simultaneously with two UHI lamps) and articulated topography. This condition affected the CWC site the most, where pixels on the western side of the image, characterised by a constant darker area, could not be strictly compared to those of the eastern side. For the coralligenous site, the presence of sunlight improved the condition, giving a more homogeneous illumination of the surveyed track, which was also favoured by a rather flat topography.

Furthermore, the low accuracy of the underwater positioning system sometimes induced image distortion and a lower spatial resolution of the geocorrected images, more evident at the CWC site. Finally, the difficulty in maintaining the correct ROV altitude in areas characterized by seafloor heterogeneity, a common trait at both study sites, may have influenced the reliability of UHI acquisition and, therefore, classification [26]. 

Based on our experience, we can summarise some best practices needed to acquire good quality hyperspectral images using an ROV. A rigorous UHI survey functional to seafloor mapping requires:an ROV ensuring a constant heading and altitude above the seafloor and suitable to host the UHI and other devices (e.g., RGB camera, lamps);an efficient positioning system for the ROV and the UHI itself, able to provide accurate and adequately dense navigation data;an appropriate lamp system to illuminate the surveyed area uniformly in function of water depth and sunlight;an RGB camera mounted vertically alike the UHI camera, to record concomitantly the seafloor for the OOI identification;an advanced background knowledge of the target area.

### 4.2. Evaluation of Spectral Libraries for Seafloor Mapping

A spectral library permits a quick and reliable classification of benthic habitats and their individual components up to taxonomic level, if a substantial number of reference spectra has been filed [7]. Its construction represents one of the most challenging and time-consuming aspects of the automatic classification of the UHI images. 

In this perspective, it appears obvious that there is a strong need to implement substantially the spectral library with respect to the deep-water scleractinians (CWCs). Living *M. oculata* appears to contain a variety of colored facies from white to pinkish hues [64]. The same holds true for *D. pertusum*, a species also present at the CWC site here considered, whose living colonies cover a chromatic spectrum from white to orange, up to reddish [65,66]. Dead skeletons of these and other CWCs are often co-occurring, further complicating the hyperspectral approach as they are characterized by whitish, yellowish and brownish colours. However, UHI is documented [5] to discriminate the optical fingerprints of white, orange and dead *D. pertusum*. 

A robust sponge taxonomic classification requires the analysis of spicules and genetics, while the external morphology or colour commonly provides only an indication of ‘morpho-species’ or ‘morpho-categories’ [67]. In the specific case of the deep-water situation here considered, the large fan-shaped sponges (i.a., *P. compressa* and *P. monilifera*, often co-occurring together with other sponges) are characterized by a wide range of colours even within the same taxon [36,68,69,70]. Therefore, collected spectra are not unequivocally associated to species, making difficult a precise taxonomic assignment.

The resilient core of the Coralligenous habitat is hard, hosting a variety of fauna and flora, the latter often seasonal and epibiontic. At times, such seasonal overgrowth may mask, often significantly but ephemerally, the underlying substrate provided by CCA and Peyssonelliales. These considerations are relevant in defining the spectral library associated with coralligenous as a unique habitat, when tested using the UHI camera. Concerning algae, supervised UHI classification was unable to map accurately different red algae species, due to their similarity on the optical fingerprint, deciding instead for grouping [26]. We tested that the spectral fingerprint of CCA and Peyssonelliales as an opting group is conspicuous enough to be distinguished in the natural environment, where their presence prevails with seasonal and accidental signals (such as green algae bloom or megabenthos). 

### 4.3. SAM Classification 

For our scope, the SAM classification is considered ideal because it is intensity independent (LED lamps and sunlight illumination) and focuses only on identifying the spectral similarity (i.e., colour). The SAM method can eliminate the effect of the spectral brightness values (i.e., spectral vector lengths in feature space) on the classification and it is insensitive to the data variance, imparting a significant advantage for the analysis of regions with complex terrain [63]. On the other hand, this method is highly dependent on the wavelength ranges and on the thresholds selected, which are arbitrary [62]. In our study, we choose several ROIs for the same benthic class for different illumination conditions, because the illumination is not influencing only the reflectance intensity, but also colour variation (e.g., colonial cnidarian 1 in Figure 2E results blue). 

At the CWC site, SAM was functional in recognizing colonial cnidarians and sponges. However, the method proved inadequate in discriminating between mud and bedrock substrates, probably characterized here by similar spectra, hampering a reliable mapping of the seafloor (Figure 4 B). 

The CCA+P and associated organisms were correctly classified at the coralligenous site, despite habitat heterogeneity. Green algae as *C. bursa* and *F. petiolata* (Green algae 1+2) appear overestimated, since SAM imported artifacts such as the build-up shadows and distortions (Figure 6). 

According to confusion matrices limited to the dataset analysed in this study, the SAM classification accuracy is higher for the CWC (84.38%) than for the coralligenous (72%) site. This result could derive from differences in habitat complexity. Firm numbers on the overall classification accuracy could be obtained by increasing the number of iterations or considering a larger dataset.

### 4.4. Evaluation of the UHI for Seabed Monitoring

Our tests document that the UHI method is able to map the habitat extent independently of the water depth and at a high level of spatial detail. The UHI provides the effective spatial coverage of CWCs habitat-formers and coralligenous builds-ups (Figure 7), which is hard to estimate using conventional methods. This level of detail is extremely useful for monitoring purposes (e.g., MSFD) enabling a quantitative and repeatable measure of habitat extent and distribution. However, this process is heavily time-consuming compared to the conventional ROV approach, mostly due to inadequate spectral libraries, which is the major limitation to date.

## 5. Conclusions

This first application of the UHI camera in the Mediterranean Sea (Southern Adriatic Sea) confirmed its potential for underwater habitat mapping in shallow and deep water.

We tested the UHI camera in two geomorphological contexts containing charismatic marine benthic habitats. We noticed that the quality of the positioning system, the illumination settings and the complexity of the seafloor affected the UHI performance and the hyperspectral image analysis. We created a preliminary spectral library for each site enabling a supervised classification (SAM), which discriminated between substrates, megafauna and megaflora in a satisfactory manner. 

Given substantially implemented spectral libraries, the UHI camera will likely represent a valid aid for habitat mapping and monitoring, in the perspective of quantifiable and repeatable classifications and European MSFD indicators.

## Figures and Tables

**Figure 1 sensors-19-02261-f001:**
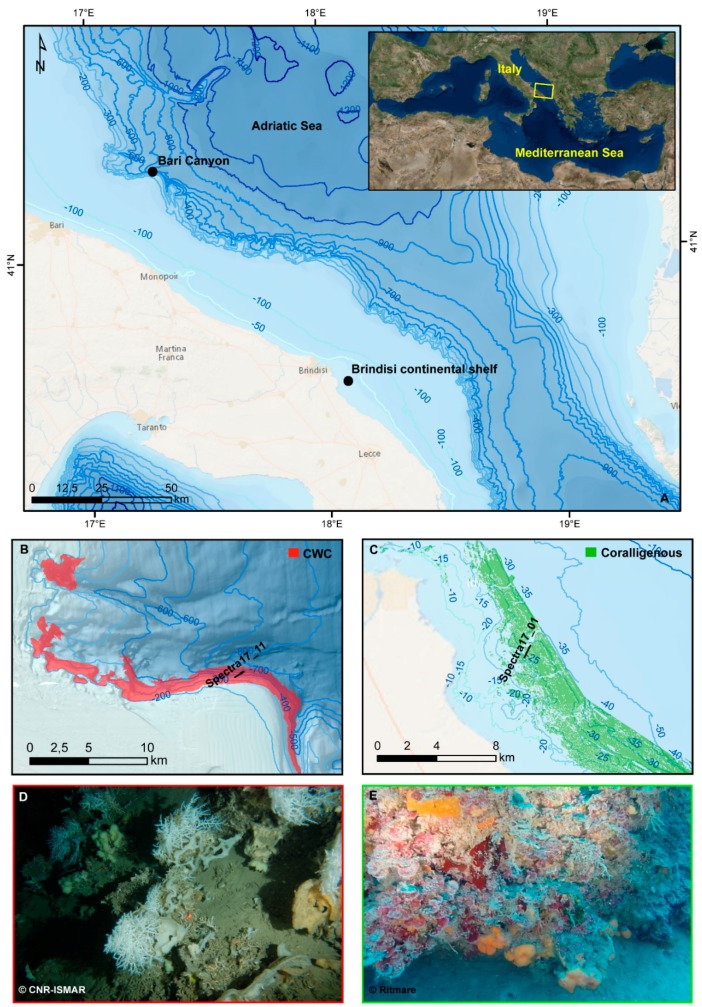
(**A**) Location of the two sites, inset shows the position in the Mediterranean Sea; (**B**) the extension of the Bari Canyon CWC province (from [56]) and (**C**) the extension of the coralligenous in the Brindisi area (black lines indicate the ROV surveys). Habitat maps produced by the BIOMAP project and further updated within the CoCoNet project. (**D**) Example of CWC habitat complexity showing colonies of *M. oculata* and large fan-shaped sponges (from [38]); (**E**) example of coralligenous characterized by CCA and Peyssonelliales, serpulids and orange encrusting sponges overprinting the calcified red algae.

**Figure 2 sensors-19-02261-f002:**
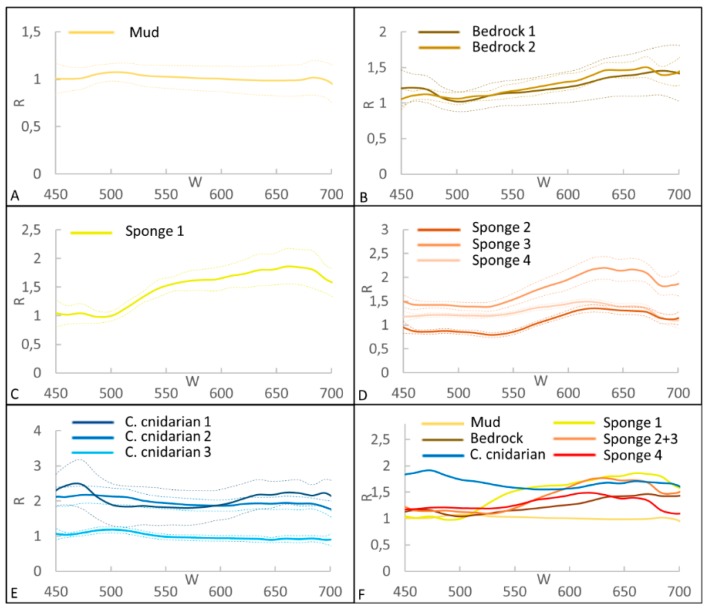
Mean spectra of the ROIs for the CWC site. R is the reflectance and W the wavelength. In **A**, **B**, **C**, **D** and **E**, dashed lines represent the standard deviation. **F** shows the synthesis of all spectra.

**Figure 3 sensors-19-02261-f003:**
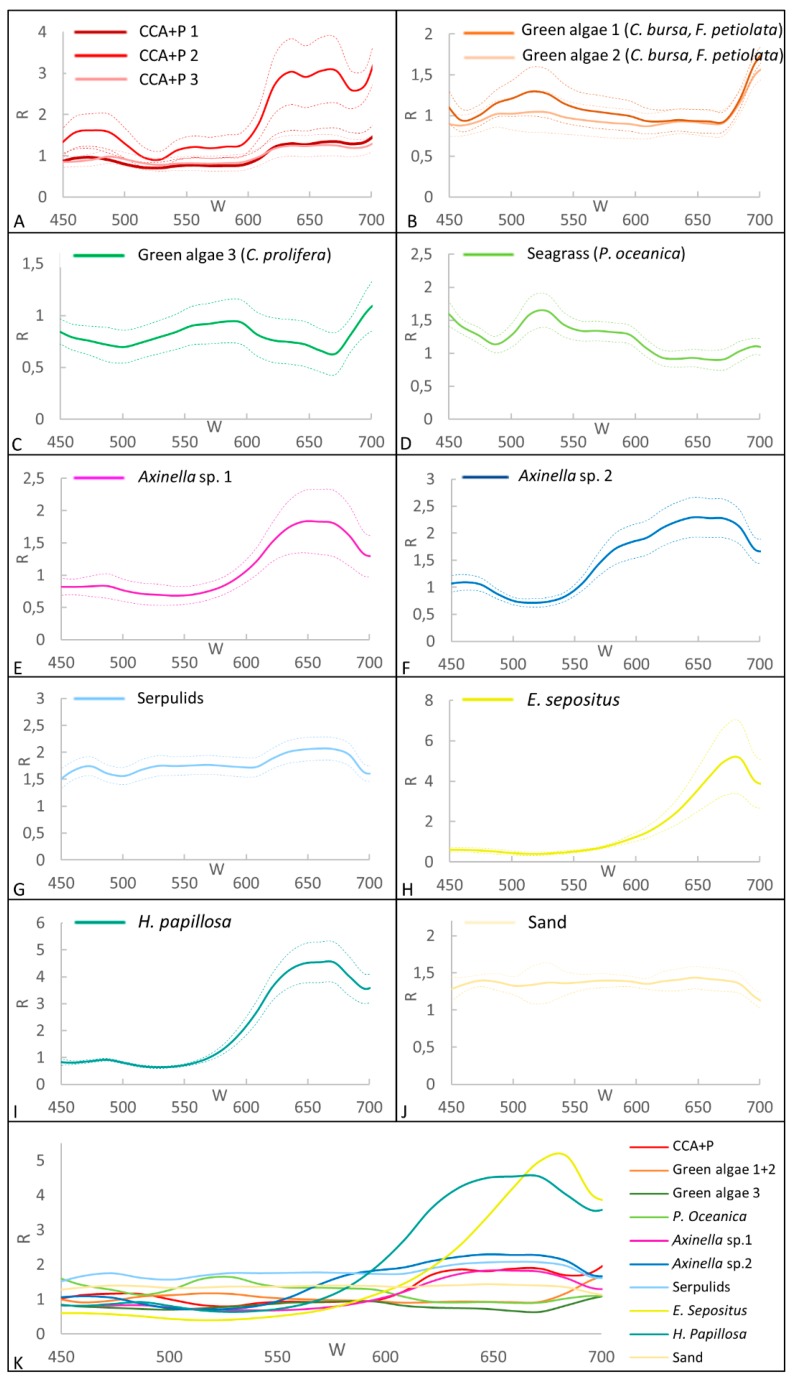
Mean spectra based on the ROIs relative to the coralligenous site. The dashed line in the graphs represents the standard deviation. **K** shows the synthesis of all spectra.

**Figure 4 sensors-19-02261-f004:**
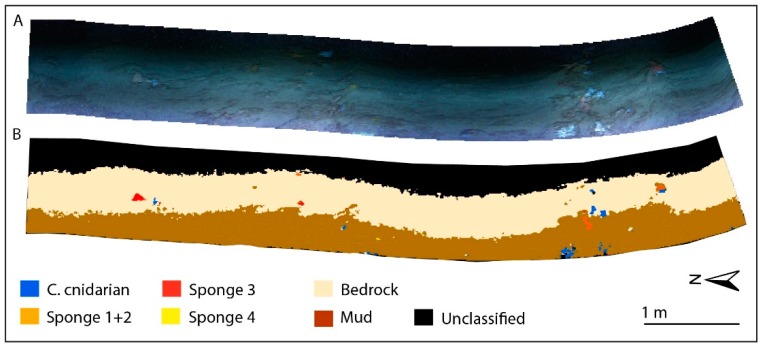
RGB UHI image of the CWC site in **A** and its SAM classification in **B**.

**Figure 5 sensors-19-02261-f005:**
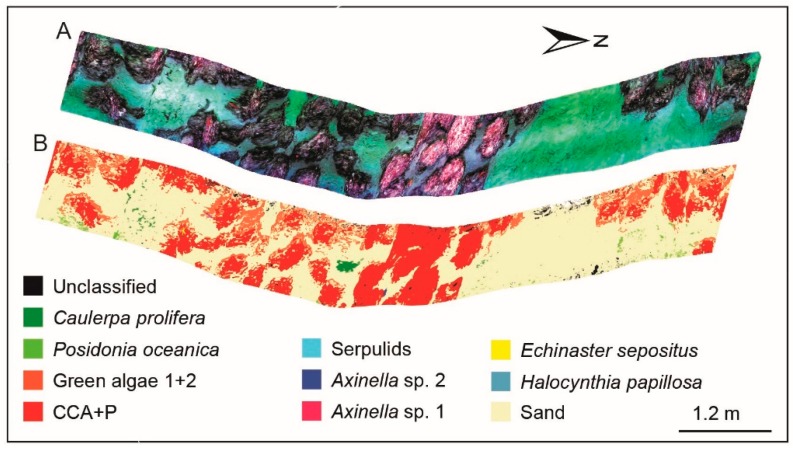
RGB UHI image of the coralligenous site in **A** and its SAM classification in **B**.

**Figure 6 sensors-19-02261-f006:**
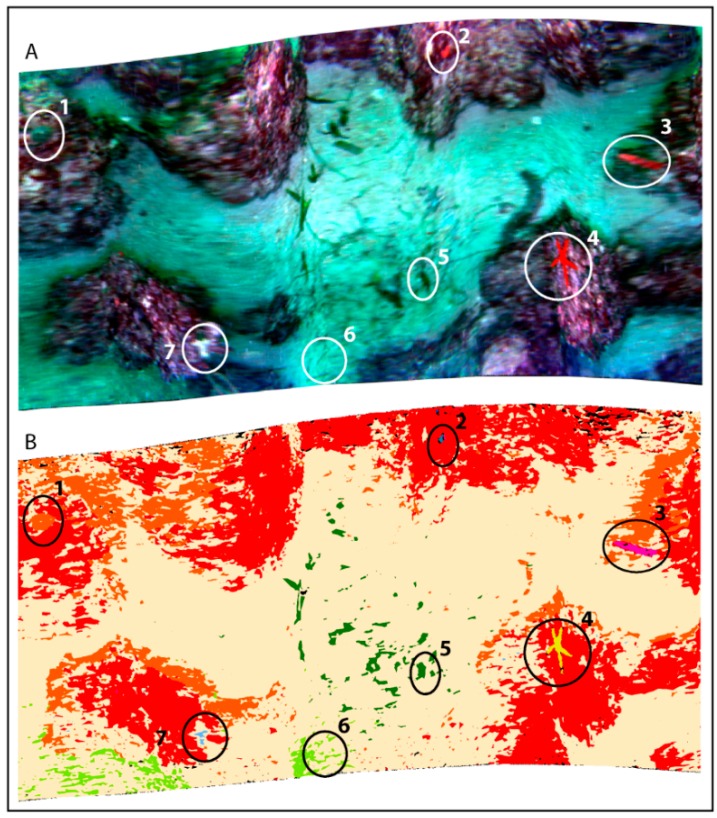
Zoom in the classified segment of coralligenous site, where numbers indicate the organism identified (**A**) and classified (**B**): 1. *C. bursa*, 2. *H. papillosa*, 3. *Axinella* sp. 1, 4. *E. sepositus*, 5. *C. prolifera*, 6. *P. oceanica*, 7. Serpulids. For the colour legend in B, see Figure 5.

**Figure 7 sensors-19-02261-f007:**
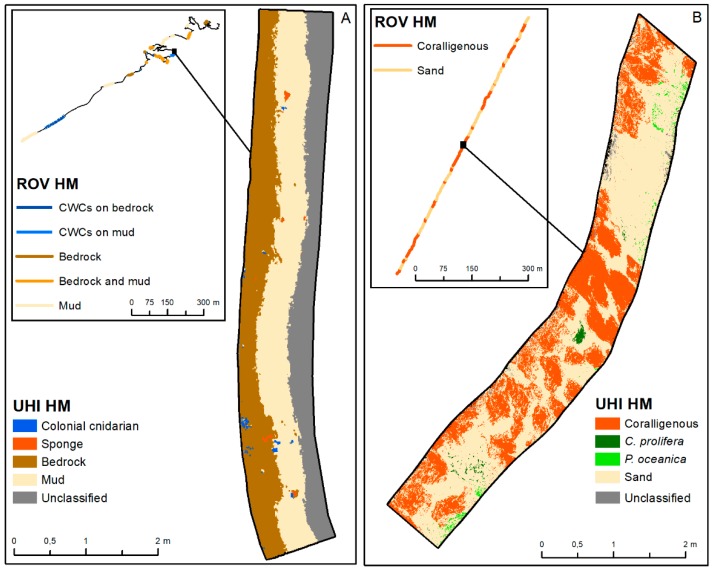
Comparison between the ROV transect classified with conventional methodologies (A and B insert) and the UHI classification for (**A**) CWC and (**B**) coralligenous sites. The image shows the higher level of detailed obtained by the UHI camera, in contrast with the larger amount of data analyzed by the ROV video.

**Table 1 sensors-19-02261-t001:** Selected ROIs for the benthic classes of the CWC site and relative threshold values used in the ‘Rule classifier’ tool.

Benthic Class	Colonial Cnidarian			Sponge				Mud	Bedrock	
ROI	1	2	3	1	2	3	4	1	1	2
**Taxonomy**	*M. oculata*/*D. pertusum*			*Hexactinellida* sp.	*Demospongiae* sp. 1	*Demospongiae* sp. 2	*Demospongiae* sp. 3			
**Threshold**	0.035	0.025	0.08	0.08	0.15	0.045	0.035	0.08	0.08	0.08

**Table 2 sensors-19-02261-t002:** Selected ROIs for the benthic classes of the coralligenous site and relative threshold values used in the ‘Rule classifier’ tool.

Benthic Class	CCA+P			Green Algae			Seagrass	Associated Organism					Sand
ROI	1	2	3	1	2	3	1	Sponge 1	Sponge 2	Serpulids	Red Starfish	Red Ascidia	1
**Taxonomy**				*C. bursa*/*F. petiolata*		*C. prolifera*	*P. oceanica*	*Axinella* sp. 1	*Axinella* sp. 2		*E. sepositus*	*H. papillosa*	
**Threshold**	0.2	0.02	0.07	0.08	0.08	0.065	0.045	0.08	0.07	0.03	0.18	0.06	0.2

**Table 3 sensors-19-02261-t003:** Confusion Matrix for the CWC site classification.

Overall Accuracy 84.38%						
	Mud	Sponge	Bedrock	C. cnidarian	TOT	UA
Mud	7	0	2	1	10	**67.78**
Sponge	0	7	0	0	7	**100.00**
Bedrock	1	1	6	0	8	**66.67**
C. cnidarian	0	0	0	7	7	**100.00**
TOT	8	8	8	8	32	
PA	**87.50**	**87.50**	**75.00**	**87.5**		

**Table 4 sensors-19-02261-t004:** Confusion Matrix for the coralligenous site classification.

Overall Accuracy 72%
	Serpulids	*E. sepositus*	*H. papillosa*	*Axinella* sp.1	*Axinella* sp.2	CCA+P	Green Algae 3	Sand	*P. oceanica*	Green Algae 1+2	TOT	UA
Serpulids	7	0	0	0	0	0	0	0	0	0	7	**100.0**
*E. sepositus*	0	8	0	0	0	0	0	0	0	0	8	**100.0**
*H. papillosa*	0	0	8	1	0	0	0	0	0	0	9	**88.9**
*Axinella* sp.1	0	0	0	6	0	0	0	0	0	0	6	**100.0**
*Axinella* sp.2	0	0	0	0	8	0	0	0	0	0	8	**100.0**
CCA+P	1	0	0	1	0	8	0	1	0	0	11	**61.5**
Green algae 3	0	0	0	0	0	0	3	0	0	0	3	**100.0**
Sand	0	0	0	0	0	0	5	5	4	1	15	**25.0**
*P. oceanica*	0	0	0	0	0	0	0	0	4	0	4	**100.0**
Green algae 1+2	0	0	0	0	0	0	0	2	0	7	9	**70.0**
TOT	8	8	8	8	8	8	8	8	8	8	80	
PA	**87.5**	**100**	**100**	**75**	**100**	**100**	**37.5**	**62.5**	**50**	**87.5**

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
