# Peer review of "Application of Hyperspectral Imaging to Underwater Habitat Mapping, Southern Adriatic Sea"

_sensors, 2019, doi:10.3390/s19102261_

Round 1

Reviewer 1 Report

General comments:

Overall this is an interesting paper with some good benthic habitat mapping results. However the paper has several shortcomings, which should be addressed before the paper could be recommended for publication.

First of all there are far too many grammatical/language errors in the paper. Those errors make it hard to follow the text and sometimes even impede understanding the text. I insist the entire paper to be critically examined and edited in order to address the grammatical errors as well as rewrite long and unclear sentences. I emphasize only some sentences in my specific comments section, that need to be rewritten.

In addition there exists some disorder in the structure of the paper. Some of the chapters contain information that should belong to some other chapter or sub-chapter. For example all general discussion about advantages/disadvantages of one or other method compared to the other method should belong to the Introduction chapter. At the same time all the technical details and instrument characteristics should be revealed in the Materials and methods chapter. I advise authors to carefully read and arrange chapters as required.

The Spectral Angle Mapper (SAM) method was chosen as a classification method in the current paper. SAM method should be relatively insensitive to illumination effects since it is invariant to absolute values of reflectance. It means that SAM method is primarily responding to spectral shape similarities and differences, not to the absolute values. As authors have experienced the illumination unevenness in their study areas, then the SAM method seemed to be a logical choice. However, in their study they have chosen several training classes from the same benthic class, but different illumination conditions. Those additional training classes seem rather unnecessary in case of SAM classification method, as spectral shapes do not depend on illumination conditions. My question would be, whether there would be any change in classification results if those additional training classes were excluded.

Specific comments:

Abstract, line 42: “The aim of the project was to evaluate the potentiality of this methodology for monitoring contrasting benthic habitats…“ Consider using ‘two different benthic habitats’ instead of ‘contrasting benthic habitats’.

Abstract, line 43: “We created two spectral signature catalogues for …“. Unclear sentence. It seems that you created one spectral library for main habitat builders and the other for different substrates, while in fact you created one library for Coralligenous habitat and the other for CWC habitat.

Abstract, line 46:  “....the UHI was able to classify the most important macro-organisms, to the lower possible taxonomic rank.“  Confusing sentence. The lowest possible taxonomic rank is species. Do you state that you managed to classify benthic organisms to species level?

Abstract, line 48: “Albeit preliminary, first results demonstrated…“ Unclear sentence, needs to be reworded.

Introduction, page 2, line 58: “Since every object absorbs and reflects light at different wavelengths, depending on its own colour and pigmentation.“ Unclear sentence, needs to be reworded.

Introduction, page 2, line 59:  “The portion of the spectrum reflected the spectral signature...“ Unclear sentence, needs to be reworded.

Introduction, page 2, line 61: “This technology has been previously applied to airborne remote sensing, both in terrestrial and marine settings“. Consider using ‘marine environment’ or ‘water environment’ instead of ‘marine settings’.

Introduction, page 2, lines 70 - 75: “The Underwater Hyperspectral Image (UHI), developed and patented by Ecotone AS…“. The specific information regarding newly developed UHI system should belong to the Materials and Methods chapter.

Introduction, page 2, line 90: “The application of UHI for mapping and monitoring marine benthic habitats...” Too long sentence, needs rewriting.

Introduction, page 2, line 95: “In the present research, we set out the results of the application of the UHI...” Too long sentence, needs rewriting.

Subchapter ‘1.1 Study area’ should belong under Materials and Methods chapter.

Study area, page 3, line 127: “...best embodied by the colonial scleractinian Madrepora oculata and subordinately Desmophyllum pertusum plus the solitary Desmophyllum dianthus…“. It’s not suitable to use the term ’plus’ in the scientific paper.

Materials and methods, page 5, lines 153-158:  “Since each UHI pixel is assigned…“. An analysis about the advantages of UHI compared to the traditional photography should belong to the Introduction chapter.

Materials and methods, page 5, line 176: “ROV ‘flight’ at constant altitude of approximately 1-1.5 m, whose results in a UHI coverage width of 1.2-1.7 m.“ Unclear sentence, needs to be reworded..

Subchapter ‘2.3 RGB image data processing’ duplicates the text of the previous subchapter.

Materials and methods, page 6, line 212: “A standard lamp with known spectral properties was used to create a map between…“ You don’t create a map; it’s either a relationship between digital numbers and radiance or gain and offset values that you need if you want to convert digital numbers to radiance.

Materials and methods, page 7, line 248: “The model was trained selecting “Region Of Interest” (ROI) that include as many spectral signatures as possible…“. Confusing sentence. Why is it stated that your ROI includes as many spectral signatures as possible, while in fact you only need spectral signatures representative of substrates present in the study area.

Materials and methods, page 7, line 252: “The ROIs were defined manually for…“. Specifications of RGB image (eg. bands, wavelengths) should belong to the subchapter – ’2.3 RGB image data processing.’

Materials and methods, page 7, line 267:  “For the CWC site, we identified 4 categories…“. Consider using 4 benthic classes instead of 4 categories.

Materials and methods, page 7, line 282:  “...we subdivided the habitat into 5 main categories…“. Consider using 5 benthic classes instead of 5 categories.

Results, page 8, line 320: “3.1. Catalogue of spectral signatures.“ Consider ’spectral library’ instead of ’catalogue of spectral signatures’.

Results, page 8, line 321: “For the CWC site, we extracted the mean spectrum from the corrected UHI image for 10 ROIs ….“ Unclear sentence, needs to be reworded..

Results, page 9, line 340: “The level of reflectance for the Colonial cnidarian spectrum varies in relation to the level of illumination of the image.“ Illumination differences affect reflectance values, not spectral shapes of OOIs. It is seen from the figure that the spectra of cnidarians 2 and 3 are similar in shape, but different in reflectance values. Therefore it may be concluded that difference is caused by illumination unevenness. At the same time the shape of reflectance spectra of cnidarians 1 differs from others. It is most probably caused by colour differences, not by illumination unevenness.

Results, page 9, line 343: “For the coralligenous site, we extracted the spectral signatures from corrected UHI image for 13 ROIs…“. Unclear sentence, needs to be reworded..

Results, page 12, line 368: “The SAM classification done with 10 ROIs identified 6 classes…“ Confusing sentence. Does it mean that 4 ROIs did not retrieve any benthic classes? Or does it mean that all ROIs retrieve benthic classes, but initial classes were later merged into 6 benthic classes?

Conclusion, page 18, line 560: “This allowed the production of a more accurate classification of the Coralligenous habitat site off Brindisi, than the one produced for the CWC habitat site in the Bari Canyon.” Authors are stating that they achieved more accurate classification results in shallower Coralligenous site, while the results of accuracy assessment (confusion matrixes) show that better classification results were acquired in deeper CWC site (overall accuracy 84,38% in CWC site and 72% in Coralligenous site).

Author Response

General comments:

Point 1: Overall this is an interesting paper with some good benthic habitat mapping results.  However the paper has several shortcomings, which should be addressed before the paper could be recommended for publication.

Response 1: ok, we amended the text accordingly.

Point 2: First of all there are far too many grammatical/language errors in the paper. Those errors make it hard to follow the text and sometimes even impede understanding the text. I insist the entire paper to be critically examined and edited in order to address the grammatical errors as well as rewrite long and unclear sentences. I emphasize only some sentences in my specific comments section, that need to be rewritten.

Response 2: ok, we completely revise the text checking grammatical/language errors in the paper.

Point 3: In addition there exists some disorder in the structure of the paper. Some of the chapters contain information that should belong to some other chapter or sub-chapter. For example all general discussion about advantages/disadvantages of one or other method compared to the other method should belong to the Introduction chapter. At the same time all the technical details and instrument characteristics should be revealed in the Materials and methods chapter. I advise authors to carefully read and arrange chapters as required.

Response 3: ok, we followed this suggestion and we changeed the structure of the paper when required.

Point 4: The Spectral Angle Mapper (SAM) method was chosen as a classification method in the current paper. SAM method should be relatively insensitive to illumination effects since it is invariant to absolute values of reflectance. It means that SAM method is primarily responding to spectral shape similarities and differences, not to the absolute values. As authors have experienced the illumination unevenness in their study areas, then the SAM method seemed to be a logical choice. However, in their study they have chosen several training classes from the same benthic class, but different illumination conditions. Those additional training classes seem rather unnecessary in case of SAM classification method, as spectral shapes do not depend on illumination conditions. My question would be, whether there would be any change in classification results if those additional training classes were excluded.

Response 4: yes, we chose the SAM method because of the illumination unevenness along transects. Nevertheless, without the additional training classes the accuracy of the two classifications was lower. Along transects (especially for the deeper site), the LED lamps modified also the colour of our OOIs according to altitude and inclination. For example considering the colonial cnidarian benthic class: the SAM used the ROIs 2 and 3 to classify the most part of the colonies, but in some cases, we needed the ROI 1 for reaching also the blue component of the colonies. ROIs 2 and 3 may seem redundant, but we used them synergistically. During the post processing (Rule classifier), we noticed that using both Colonial cnidarian 2 and 3 (same shape, different intensity), we are able to reach a greater number of useful pixels.

Specific comments:

Point 1: Abstract, line 42: “The aim of the project was to evaluate the potentiality of this methodology for monitoring contrasting benthic habitats…“ Consider using ‘two different benthic habitats’ instead of ‘contrasting benthic habitats’.

Response 1: ok, we changed this sentence (lines 41-42).

Point 2: Abstract, line 43: “We created two spectral signature catalogues for …“. Unclear sentence. It seems that you created one spectral library for main habitat builders and the other for different substrates, while in fact you created one library for Coralligenous habitat and the other for CWC habitat.

Response 2: yes, we created two spectral libraries, one for Coralligenous habitat and the other for CWC habitat. We rewrote the sentence: “We created a spectral signature library for each site, considering the main habitat builders and different substrates.”

Point 3: Abstract, line 46:  “....the UHI was able to classify the most important macro-organisms, to the lower possible taxonomic rank.“  Confusing sentence. The lowest possible taxonomic rank is species. Do you state that you managed to classify benthic organisms to species level?

Response 3: yes, the lowest possible taxonomic rank is species. We rewrote the sentence “We created a spectral library for each site, considering the different substrates and the main OOI, reaching where possible, the lower taxonomic rank.” (lines 42-44).

Point 4: Abstract, line 48: “Albeit preliminary, first results demonstrated…“ Unclear sentence, needs to be reworded.

Response 4: ok, we reworded the sentence: “Despite some technical problems, first results demonstrated the suitability of the UHI camera to contribute to habitat mapping and seabed monitoring, through the achievement of quantifiable and repeatable classifications.” (lines 45-48)

Point 5: Introduction, page 2, line 58: “Since every object absorbs and reflects light at different wavelengths, depending on its own colour and pigmentation.“ Unclear sentence, needs to be reworded.

Response 5: ok, we rewrote the entire paragraph, this concept is from line 65 to 68.

Point 6: Introduction, page 2, line 59:  “The portion of the spectrum reflected the spectral signature...“ Unclear sentence, needs to be reworded.

Response 6: ok, we rewrote the entire paragraph, this concept is from line 65 to 68.

Point 7: Introduction, page 2, line 61: “This technology has been previously applied to airborne remote sensing, both in terrestrial and marine settings“. Consider using ‘marine environment’ or ‘water environment’ instead of ‘marine settings’.

Response 7: ok, used “marine environment” instead of “marine settings” (line 74).

Point 8: Introduction, page 2, lines 70 - 75: “The Underwater Hyperspectral Image (UHI), developed and patented by Ecotone AS…“. The specific information regarding newly developed UHI system should belong to the Materials and Methods chapter.

Response 8: ok, we included this information in the Materials and Methods chapter (line 131).

Point 9: Introduction, page 2, line 90: “The application of UHI for mapping and monitoring marine benthic habitats...” Too long sentence, needs rewriting.

Response 9: ok, we rewrote the entire paragraph, this concept is from line 86 to 91.

Point 10: Introduction, page 2, line 95: “In the present research, we set out the results of the application of the UHI...” Too long sentence, needs rewriting.

Response 10: ok, we rewrote the entire paragraph, this concept is from line 92 to 101.

Point 11: Subchapter ‘1.1 Study area’ should belong under Materials and Methods chapter.

Response 11: ok, we moved this subchapter in Materials and Methods chapter (lines 103-122).

Point 12: Study area, page 3, line 127: “...best embodied by the colonial scleractinian Madrepora oculata and subordinately Desmophyllum pertusum plus the solitary Desmophyllum dianthus…“. It’s not suitable to use the term ’plus’ in the scientific paper.

Response 12: ok, we replaced it.

Point 13: Materials and methods, page 5, lines 153-158: “Since each UHI pixel is assigned…“. An analysis about the advantages of UHI compared to the traditional photography should belong to the Introduction chapter.

Response 13: ok, we moved this sentence in the Introduction chapter (line 68).

Point 14: Materials and methods, page 5, line 176: “ROV ‘flight’ at constant altitude of approximately 1-1.5 m, whose results in a UHI coverage width of 1.2-1.7 m.“ Unclear sentence, needs to be reworded..

Response 14: ok, we reworded the sentence (line 139).

Point 15: Subchapter ‘2.3 RGB image data processing’ duplicates the text of the previous subchapter.

Response 15: yes, we replaced the content at this subchapter (lines 165-172).

Point 16: Materials and methods, page 6, line 212: “A standard lamp with known spectral properties was used to create a map between…“ You don’t create a map; it’s either a relationship between digital numbers and radiance or gain and offset values that you need if you want to convert digital numbers to radiance.

Response 16: Ok we rewrote the sentence, we did not create a map but we found the ratio between observed digital counts of intensity for each wave band on the sensor (line 183).

Point 17: Materials and methods, page 7, line 248: “The model was trained selecting “Region Of Interest” (ROI) that include as many spectral signatures as possible…“. Confusing sentence. Why is it stated that your ROI includes as many spectral signatures as possible, while in fact you only need spectral signatures representative of substrates present in the study area.

Response 17: yes, the sentence was unclear. We reworded in “In particular, for the CWC site, we used 10 ROIs to train the SAM classification…” (line 225).

Point 18: Materials and methods, page 7, line 252: “The ROIs were defined manually for…“. Specifications of RGB image (eg. bands, wavelengths) should belong to the subchapter – ’2.3 RGB image data processing.’

Response 18: we eliminated this sentence because of its redundancy.

Point 19: Materials and methods, page 7, line 267:  “For the CWC site, we identified 4 categories…“. Consider using 4 benthic classes instead of 4 categories.

Response 19: ok, we used “benthic classes” instead if “categories”.

Point 20: Materials and methods, page 7, line 282:  “...we subdivided the habitat into 5 main categories…“. Consider using 5 benthic classes instead of 5 categories.

Response 20: ok, we used “benthic classes” instead of “categories”.

Point 21: Results, page 8, line 320: “3.1. Catalogue of spectral signatures.“ Consider ’spectral library’ instead of ’catalogue of spectral signatures’.

Response 21: ok, we used “spectral library” instead of “catalogue of spectral signatures”.

Point 22: Results, page 8, line 321: “For the CWC site, we extracted the mean spectrum from the corrected UHI image for 10 ROIs ….“ Unclear sentence, needs to be reworded..

Response 22: we simplified the sentence in “The mean spectra of the 10 ROIs selected for the CWC site are shown in Figure 2” (line 262).

Point 23: Results, page 9, line 340: “The level of reflectance for the Colonial cnidarian spectrum varies in relation to the level of illumination of the image.“ Illumination differences affect reflectance values, not spectral shapes of OOIs. It is seen from the figure that the spectra of cnidarians 2 and 3 are similar in shape, but different in reflectance values. Therefore it may be concluded that difference is caused by illumination unevenness. At the same time the shape of reflectance spectra of cnidarians 1 differs from others. It is most probably caused by colour differences, not by illumination unevenness.

Response 23: yes, Colonial cnidarian 2 and 3 have the same shape and their difference in reflectance value is probably due to illumination unevenness. While, Colonial cnidarian 1 is different in shape showing a peak in the blue part of the spectrum. We know that colonial cnidarians can’t be blue, so we considered this colour an artefact (probably due to the lamps). For this reason, we represented the three ROIs in a unique class named “Colonial cnidarians” in the final classification.

Point 24: Results, page 9, line 343: “For the coralligenous site, we extracted the spectral signatures from corrected UHI image for 13 ROIs…“. Unclear sentence, needs to be reworded..

Response 24: we reworded the sentence (line 284).

Point 25: Results, page 12, line 368: “The SAM classification done with 10 ROIs identified 6 classes…“ Confusing sentence. Does it mean that 4 ROIs did not retrieve any benthic classes? Or does it mean that all ROIs retrieve benthic classes, but initial classes were later merged into 6 benthic classes?

Response 25: it means that all ROIs are used to perform the SAM, but in the final classification map some ROIs are merged according to their spectral similarity (see subchapter 3.1 and 3.2). We hope this new version better explain the relation between the initial ROIs and the final classification map.

Point 26: Conclusion, page 18, line 560: “This allowed the production of a more accurate classification of the Coralligenous habitat site off Brindisi, than the one produced for the CWC habitat site in the Bari Canyon.” Authors are stating that they achieved more accurate classification results in shallower Coralligenous site, while the results of accuracy assessment (confusion matrixes) show that better classification results were acquired in deeper CWC site (overall accuracy 84,38% in CWC site and 72% in Coralligenous site).

Response 26: the conclusion paragraph has been rewritten (lines 447-456).

Reviewer 2 Report

I firstly concern the novelty of this paper. The authors said they used traditional methods (SAM), and did a general classification mission, but only for different study areas. So, it is not enough for me. Even, you never show the research gap, in order to present the necessity of this study.

Specifically, why SAM was selected? Too much superior supervised classifiers should be considered, e.g., RF, DL.

For training, why only 10 ROI? .For my experience, it is really too little. For supervised classification, the training samples are very important. You need to explain why these samples were selected. (it should be related to number and types)

The writing is really bad, and it is so difficult to follow the authors and understand.

Even, Section 2.2 and Section 2.3 are the same, but only the captions. How do you submit to the journal by current shape?  

Overall, I think the current version is not a rigorous research paper, whatever the writing or experiments. So, I have to require the authors for more change to submit. 

Author Response

General comments:

Point 1: I firstly concern the novelty of this paper. The authors said they used traditional methods (SAM), and did a general classification mission, but only for different study areas. So, it is not enough for me. Even, you never show the research gap, in order to present the necessity of this study.

Response 1: We rewrote the introduction, part of the discussions and the conclusions in order to better clarify the novelty of the work. This paper describes for the first time the application of the UHI in the Mediterranean sea, evaluating its potential for mapping the extent and distribution of two key habitats in the South Adriatic Sea, in a quantifiable and repeatable manner, compliant with the Marine Strategy Framework Directive (MSFD) We highlighted  the need of new approaches to fulfil the MSFD requirements. We better describe the relevance of the selected sites and the aim of the paper in the introduction from line 53-63.

Point 2: Specifically, why SAM was selected? Too much superior supervised classifiers should be considered, e.g., RF, DL.

Response 2: in underwater inspections, the SAM seems to be the most logical choice because of its independence from illumination intensity variation. Surely, a great number of other classification tools could be considered and testing in another ad hoc paper.  The aim of the present work is the application of the Underwater Hyperspectral Imager testing the potentiality of this tool to address habitat mapping and monitoring issues.

Point 3: For training, why only 10 ROI? .For my experience, it is really too little. For supervised classification, the training samples are very important. You need to explain why these samples were selected. (it should be related to number and types).

Response 3: another reason to choose the SAM method is that: there is no specific requirement for a large amount of training samples [63]. During our processing, we tested different combinations and finally we chose the type of ROIs that better covered the diversity along the segment and the minimum number giving acceptable results.

Point 4: The writing is really bad, and it is so difficult to follow the authors and understand.

Response 4: ok, we completely checked the text and rewrote most of the paragraphs.

Point 5: Even, Section 2.2 and Section 2.3 are the same, but only the captions. How do you submit to the journal by current shape?

Response 5: ok, it was a mistake during the submission.

Point 6: Overall, I think the current version is not a rigorous research paper, whatever the writing or experiments. So, I have to require the authors for more change to submit.

Response 6: ok, we rewrote most of the paper trying to be more rigorous as required.

Round 2

Reviewer 1 Report

Current revised version of the paper is considerably better compared to the previous version. Nevertheless it seems that authors have been in a hurry resubmitting the article and text still needs extensive English editing. I still found it rather difficult to follow the text and it is the biggest shortcoming of the current paper. I also want to throw the attention to the use of past ant present tenses.

I suggest authors to take some additional time to imporove the text in the way that it was easier to follow.  

Author Response

Thank you for considerations. We asked to all the authors an effort in this sense

and we revised the text focusing on this aspect. Then, we also involved a

native speaker in this process. We hope this version is easier to follow according to your

suggestions and the paper could be considered now acceptable. 

Reviewer 2 Report

Thank you for your careful revision. Based on my concerns, I can accept your explanation, why you select SAM, and your contribution to this application.

However, I think your experiments are really biased at the process of the selection of samples. You said that “we tested different combinations and finally we chose the type of ROIs that better covered the diversity”, I am so sorry I can not agree with this experiment strategy. Because we can not always get the right combination of samples at the practice, and your results are really accidental. So, I suggest you should get samples randomly, and repeat the classification to show the average of classification accuracy. This will make your study more reliable to support your evaluation for the mapping potential. I am not sure if you can accept that, but this is really important for this type of study. Because samples always affect the results, you need trying to avoid this disturb.

At least, I think you should change that for the over accuracy at section 3.5, to support the conclusion that SAM has high overall accuracy. Only one-time classification is not enough.

The above change may require a lot of work. So, I suggest a major revision.

Author Response

This is a good point, we accept the concept that only one-time classification might not be enough to conclude on high overall accuracy. However, the idea to select randomly the samples is a good option in case of even settings throughout the hyperspectral image, but as we explained in the paper, the underwater hyperspectral images collected in our study show fluctuations in altitude and illumination. This is a common issue collecting UHI data. Knowing this, we consider acceptable to carefully choose the ROIs based on visual expert interpretation. In order to increase the dataset for the accuracy evaluation, we evaluated also the possibility to use these ROIs to classify other hyperspectral images available from the sites, but the time made available for revision is not enough to process new data and include it in this paper. So in the end, we made a change in the 4.3 subchapter (lines 447-450) underling that firm numbers on the classification accuracy could be reached enlarging the dataset or increasing the number of iterations. However in order to indicate the potential of the method, the classification accuracy on the limited dataset used in this study, is given in Table 2 and 3.

We hope you will find this acceptable for the paper.

Round 3

Reviewer 2 Report

I don't have any further comments. Thank you